# Mechanistic Actions of microRNAs in Diabetic Wound Healing

**DOI:** 10.3390/cells9102228

**Published:** 2020-10-02

**Authors:** Marija Petkovic, Anja Elaine Sørensen, Ermelindo Carreira Leal, Eugenia Carvalho, Louise Torp Dalgaard

**Affiliations:** 1Department of Science and Environment, Roskilde University, 4000 Roskilde, Denmark; elaine@ruc.dk (A.E.S.); ltd@ruc.dk (L.T.D.); 2Center for Neuroscience and Cell Biology, University of Coimbra, 3004-504 Coimbra, Portugal; ecleal@cnc.uc.pt (E.C.L.); ecarvalh@cnc.uc.pt (E.C.); 3Institute for Interdisciplinary Research, University of Coimbra, 3030-789 Coimbra, Portugal; 4Department of Geriatrics, University of Arkansas for Medical Sciences, and Arkansas Children’s Research Institute, Little Rock, AR 72205, USA

**Keywords:** diabetic wound healing, chronic wounds, inflammation, proliferation, remodeling, tissue and circulating microRNAs, tissue regeneration

## Abstract

Wound healing is a complex biological process that is impaired under diabetes conditions. Chronic non-healing wounds in diabetes are some of the most expensive healthcare expenditures worldwide. Early diagnosis and efficacious treatment strategies are needed. microRNAs (miRNAs), a class of 18–25 nucleotide long RNAs, are important regulatory molecules involved in gene expression regulation and in the repression of translation, controlling protein expression in health and disease. Recently, miRNAs have emerged as critical players in impaired wound healing and could be targets for potential therapies for non-healing wounds. Here, we review and discuss the mechanistic background of miRNA actions in chronic wounds that can shed the light on their utilization as specific wound healing biomarkers.

## 1. Introduction

Healthy skin is crucial for human body integrity. It is the largest organ system, protecting against mechanical forces, fluid imbalance, thermal dysregulation, and infections [1]. Over time, diabetes induces critical alterations in the skin, including a dysfunctional ability for skin regeneration after a wound [2], which can be partially attributed to insulin resistance [3].

Often complicated by underlying conditions such as long-standing diabetes, vascular disease or associated neuropathy, acute wounds can adopt the clinical phenotypes of chronic wounds [4]. Non-healing wounds share the same features, such as a lack of oxygen and nutrient supply and microbial contamination, delaying the wound healing progress [5]. Among the major chronic wound etiologies, including venous and pressure ulcers, diabetic foot ulceration (DFU) is one of the most frequent complication of diabetes, occurring in about 20% of the diabetic population [2]. Due to required hospitalizations and amputations for their management, they represent a serious public health problem. While about 80% of DFUs heal over a 1-year follow-up, including those that undergo minor amputation, about 60% of DFUs are infected at the time of first clinical presentation. Integrated multidisciplinary treatments are necessary and include frequent follow-up for wound clean-up and debridement, antibiotics treatment, pressure relief by off-loading, while, in severe cases, non-healing ulcers require hospitalizations and major limb amputation [6].

In healthy skin, injury triggers acute inflammatory responses beginning with the recruitment of neutrophils, monocytes, and mast cells to the site of injury. Besides significant alteration in the expression of inflammatory cytokines, such as Tumor Necrosis Factor (TNF)-α, Interleukin (IL)-6, IL-8 and growth factors that coordinate wound repair [7], diabetes can also cause important alterations in skin insulin action, which is crucial for skin glucose and lipid homeostasis [8]. It has been documented that hyperglycemia changes the profile and action of microRNAs (miRNA). Importantly, insulin resistance, an underlying cause of type 2 diabetes (T2D) development, can be present long before any alteration in the circulating glucose is detected, especially in high-risk subjects [9]. This indicates that hyperinsulinemia may be present even under normal glucose conditions, likely causing significant alterations at the vascular level, long before diabetes is diagnosed. High insulin levels have also been shown to modulate miRNA expression and activation [10,11]. This suggests that miRNA expression and function may be altered long before hyperglycemia is diagnosed, causing important changes in metabolism from an early stage. Furthermore, microRNAs (miRNAs) are not only important epigenetic regulators of metabolism [12], but the expression of these key molecules can also be highly regulated by epigenetic modifications, including histone modifications and DNA methylation [13,14,15]. Therefore, it is important to uncover these early changes in miRNA expression in order to best target for treatment those at the highest risk. The purpose of this review is to chart the molecular mechanisms by which miRNAs are involved in mediating the diabetes-induced impairment in wound healing.

## 2. The Dynamic Phases of Normal Wound Healing

A cutaneous wound alters the epithelial integrity of the skin. Partial-thickness cutaneous wounds (e.g., superficial lesions and abrasions) heal with the aid of the underlying epithelium providing the epidermal cells that migrate and proliferate, as well as skin appendages (i.e., hair follicles and surrounding skin glands). On the other hand, full-thickness cutaneous wounds heal by going through three coordinated and interrelated phases. Wound healing is a complex and dynamic physiological process, arbitrarily divided into hemostasis and inflammation, proliferation, and remodeling that take place inside the wound and surrounding tissue [2]. In contrast to acute wounds that progress through the various phases of wound healing linearly in healthy individuals, diabetic wounds become stalled and progression does not occur in synchrony due to diabetes-associated neuropathy and dysfunction of the immune system [16].

### 2.1. Hemostasis and the Inflammatory Phase

The first phase of wound healing begins immediately upon injury and is completed within hours. It is dedicated to hemostasis and the formation of a provisional wound matrix. The normal progression of this initial short phase is critical for the subsequent steps that occur during the ensuing phases of wound healing [17]. Wounding represents trauma to blood vessels, resulting in hemorrhage. The immediate response of the injured endothelial cellular membrane is the peripheral vasoconstriction that limits bleeding while exhausting the surrounding tissues of oxygen and nutrient supply, carried by the blood. Platelet activation triggers the coagulation cascade, in order to form the blood clot that seals the damaged endothelium (Table 1) [18]. 

A significant amount of evidence supports that macrophages have an important role in the early inflammatory phase mediating the proper healing. In particular, dermal macrophages are settled near the hair appendages impacting hair growth via macrophage-specific WNT-signaling [19]. They either respond to injury by distinguishing damage-associated molecular patterns (DAMPs) boosting a strong pro-inflammatory cascade, or, in the case of infection, recognize pathogen-associated molecular patterns (PAMPs) [20]. While dermal macrophages pioneer the immediate local response, circulating monocytes originating from the bone marrow are recruited to the wound site and upon stimulation quickly differentiate into macrophages [21]. The phenotype of macrophages is rather complex and takes into account many parameters, such as surface markers, their function described through cytokines, growth factors as well as chemokine content they secrete as summarized in the recent review by Krzyszczyk and colleagues [22]. The most common classification distinguishes the classically activated, pro-inflammatory or M1 macrophages and the alternatively activated, anti-inflammatory macrophages (pro-regenerative) or M2 macrophages [21]. However, they are not distinct groups but rather form an interchangeable phenotype spectrum [22]. The pro-inflammatory M1 macrophages phagocyte bacteria and debris and are important in the beginning of the inflammatory phase and they differentiate into M2 macrophages which release anti-inflammatory cytokines that are responsible for resuming this phase, paving the way to the proliferative phase [23]. Moreover, neutrophils are highly active during the repair of cutaneous wounds; they are mobilized to the wound site immediately after the injury by diapedesis (Table 1). Their main function is to phagocyte debris and bacteria, and eliminate them by generating highly reactive oxygen species (ROS), proteases and pro-inflammatory cytokines [24].

Lymphocytes are also known to play an important role in wound healing (Table 1) [24,25]. The lack of lymphocytes was found to impair the inflammatory response during injury with a consequence of impaired angiogenesis and an increase in metalloproteinase levels [24]. In addition, several subsets of T-cell [25], B-cell [26], Natural Killer (NK)T-cell [27] and regulatory T-cells [28] contribute to wound healing (Table 1) (Figure 1).

### 2.2. The Proliferative Phase

The proliferative phase follows up on the subsiding inflammation. This phase occurs 4–21 days following injury, in the wound microenvironment. At this point, the cellular mechanisms predominantly support the repair of the damaged skin tissue (fibroplasia), the reestablishment of the blood vessel network (angiogenesis) and the impermeable barrier (re-epithelialization) (Table 1) [29]. This phase is characterized by high cellular activity due to the increased keratinocyte migration and proliferation at the wound edge to cover the wound site, where these cells attach to the basement membrane [30]. As the center of the wound is relatively avascular, restoration of the vascular network begins in the early inflammatory phase when transforming growth factor-β1 (TGF-β1), platelet-derived growth factor (PDGF) and fibroblast growth factor (FGF) are initially secreted by activated platelets [31] along with vascular endothelial growth factor (VEGF) and hypoxia-inducible factor-1 (HIF-1), which is triggered by hypoxia and the ischemic environment [32]. Following injury, proliferating fibroblasts migrate in high abundance to the wound site, predominantly due to their stimulation by TGF-β and PDGF. In addition, they secrete extra cellular matrix (ECM) proteins (hyaluronan, fibronectins and proteoglycans) and subsequently produce collagen and fibronectin [33]. The provisional matrix, which replaces the clot at the site of a wound, is termed granulation tissue, which fills the wound space providing the scaffolding spot for migrating fibroblasts that differentiate into myofibroblasts and are responsible for collagen production at the wound site (Figure 1) [34].

### 2.3. The Remodeling Phase

A visibly closed wound is the beginning of the final healing phase. As the wound closes, the immature fibrin matrix and granulation tissue are replaced by collagen and the scar is formed. By exerting tension on the matrix, fibroblasts are able to organize collagen fibrils into sheets and cables, therefore influencing the alignment of the collagen fibers [2]. The remodeling phase is the maturation of the scar tissue, where immature collagen type III replaces the mature type I collagen [35]. Collagen remodeling during the transition from granulation tissue to scar is governed by proteolytic enzymes called matrix metalloproteinases (MMPs), which are secreted by skin-cell-like macrophages, epidermal cells, and endothelial cells, as well as fibroblasts [36]. Wound remodeling continues for up to 2 years, during which time there is no net increase in the collagen content. On the other hand, the collagen fibers spatially rearrange into a more organized reticular network through cross-linking, coordinated by the local soluble factors, giving the scar its tensile strength (Figure 1) [37].

## 3. Impairments Observed in Wound Healing in Long-Standing Diabetes

The prolonged exposure to hyperinsulinemia and/or hyperglycemia is associated with damage, dysfunction, and failure of several organs, including the skin. One of the most severe and debilitating complications of diabetes is the development of chronic, non-healing DFU, for which the underlying mechanisms are still poorly understood.

### 3.1. Persistent Inflammation in the Non-Healing Wound Environment

In part due to the diabetes-associated persistent low-grade inflammation, peripheral vascular disease and peripheral neuropathy, healing and tissue repair are impaired under diabetic conditions promoting the occurrence of chronic non-healing wounds [59,60].

Chronic diabetic wounds are commonly characterized by persistent inflammation, the presence of infection and biofilms, senescent fibroblasts, highly proliferative although non-migratory keratinocytes, poor angiogenesis, reduced production of extracellular matrix, excessive proteolytic enzymes and an increase in reactive oxygen species (ROS) [61,62,63]. Reduced blood flow restricts the migration of leukocytes [25], keratinocytes, fibroblasts, and endothelial progenitor cells to the wound site [64].

In addition, non-healing wounds fail to mount an acute inflammation response, which leads to a persistent inflammatory state that contributes to poor healing [65,66]. Exudates of chronic wounds show the presence of elevated levels of pro-inflammatory mediators [67,68]. In addition, an excessive infiltration of inflammatory cells involving neutrophil granulocytes and pro-inflammatory M1 macrophages further support the pro-inflammatory environment in chronic wounds (Figure 1) [56,67,68,69].

### 3.2. Altered Macrophage Polarization in DFUs

Dysregulation of macrophage function is one of the primary pathologies associated with diabetic wounds [70]. The inflammatory phase is characterized by an acute inflammatory response by M1 macrophages. M1 macrophages produce nitric oxide (NO), ROS, IL1 and IL6 and TNFα, favoring an inflammatory environment. Additionally, they also produce MMP2 and MMP9 in order to disassemble the extracellular matrix and clear the way for the influx of inflammatory cells [71]. At a later stage in the inflammatory phase, polarization of macrophages towards the M2 phenotype is important in order to promote tissue regeneration events such as proliferation, granulation tissue formation, and angiogenesis. M2 macrophages provide the healing environment with the release of growth factors such as PDGF, IGF1, VEGF and TGFβ1, as well as the inhibitor of metalloproteinases 1 (TIMP1) to allow the ECM synthesis [21]. Thus, the resolution of inflammation requires the phenotypic shift from the pro- to the anti-inflammatory state as the healing progresses.

Under diabetic conditions, macrophages present a deficient phagocytic capacity, and are not able to phagocyte apoptotic cells, such as apoptotic neutrophils, resulting in an imbalanced inflammatory status with higher levels of pro-inflammatory cytokines and lower levels of anti-inflammatory mediators [72,73]. Pro-inflammatory macrophage depletion in the early stages of inflammation can also lead to aberrant angiogenesis, poorly structured granulation tissue and a defect scar in the subsequent phases of healing [74].

Moreover, M2 macrophages produce fibrogenic TGF-β1 growth factors, which can enhance the transformation of fibroblasts to myofibroblasts, resulting in hypertrophic scaring [75]. In addition, clearance of the circulating macrophages from the systemic circulation, 1–2 weeks after human skin transplantation, in the human hypertrophic scaring nude mouse model, improved scar formation and collagen remodeling [76].

This suggests an important role for the dynamic and reversible phenotypic changes of macrophages during this phase of wound healing. The pro-inflammatory level, in diabetic conditions, with the contribution of dysfunctional macrophages leads to a deficient transition from the M1 to the M2 macrophage phenotype in chronic wounds which delays the inflammatory phase and impairs wound healing (Figure 1) [77,78].

### 3.3. Excessive Tissue Damage in Non-Healing Wounds

The accumulation of inflammatory cells in non-healing wounds lead to increased production of proteases that degrade the ECM further damaging the tissue [79]. Both the expression and activity of MMPs are increased in diabetic chronic wounds when compared to normal acute wounds, whereas the expression of endogenous tissue inhibitors of MMPs is reduced [80,81,82]. Therefore, an increase in the proteolytic activity observed in diabetic wounds contributes to the poor formation of new connective tissue, further impairing the healing process. Moreover, premature senescence due to constant exposure to an inflammatory environment and elevated oxidative stress has been found in cells in chronic wounds (Figure 1) [83].

## 4. microRNAs

### 4.1. Biogenesis

In recent years, microRNAs (miRNAs) have gained interest as the key to understanding not only cellular evolutionary and developmental processes, but also due to their association with different cellular differentiation states [84,85]. They have also been described as important biomarkers in more complex metabolic diseases [85,86,87]. The small endogenous miRNAs, ~22 nucleotides in length, belong to a class of non-coding RNAs first discovered in 1993 in the nematode *Caenorhabditis elegans* in which a short RNA from the lin-4 gene repressed the mRNA of another gene (the lin-14 gene) [88,89]. MiRNAs were later also detected in vertebrates [90], and since then, the list of known mature miRNAs in humans has expanded and now counts more than 2600 different sequences [91]. MiRNAs predominately function as post-transcriptional gene regulators. While each miRNA is predicted to have multiple potential target messenger RNAs (mRNAs), a single gene can be modulated by several miRNAs [92], hence increasing the complexity by which miRNAs can fine-tune gene expression.

The first step in the biogenesis of miRNAs is the generation of a primary (pri) miRNA transcript (pri-miR) (Figure 2). MiRNAs are primarily transcribed in an RNA-polymerase II dependent manner driven by their own promotors or shared with the host genes if the miRNA resides within an intron of a coding sequence [93]. Thus, pri-miRs contain a 5′cap and are 3′polyadenylated but contain one or more characteristic imperfect stem-loop hairpin structures [94]. Pri-miRs are processed within the nucleus by the microprocessor complex, consisting of a ribonuclease III, DROSHA and its associated cofactor DiGeorge critical region 8 gene (DGCR8) to form a ~70nt precursor (pre) miRNA (pre-miR) having a 2 nucleotide overhang at the 3′ end [93]. A nuclear pore complex consisting of Exportin 5 (XPO5) and GTP-binding nuclear protein ranGTP recognizes this overhang and transports the pre-miR to the cytoplasm [95,96]. Another RNase III enzyme, DICER, recognizes the 3′ overhang and the hairpin structure to cleave off the loop of the pre-miR, while leaving the mature miRNA in a duplex form.

The mature miRNA duplex is loaded into the Argonaute RISC Catalytic Component 2 (AGO2), which constitutes the RNA silencing complex (RISC) together with trans-activation response RNA-binding protein (TRBP), and protein activator of interferon-induced protein kinase EIF2AK2 (PRKRA). Subsequently, the “passenger” strand of the miRNA duplex is degraded [97,98]. Selection of the mature single-stranded miRNA and removal of the passenger strand depends on the relative thermodynamic stability of the two ends of the small RNA duplex with a lower stability favoring selection [99]. Yet, the passenger strand can also be active in silencing although most often to a lesser extent than the mature major strand miRNA [100]. Throughout this review, we will use the short names of miRNAs that only exist in one form, omitting the 5′ or 3′ end extension of the miRNA name. The systematic names of the miRNAs are available at www.mirbase.org [91].

The active RISC complex loaded with the mature miRNA identifies possible target mRNAs through sequence complementary of the miRNA seed sequence to the 3′ untranslated region (UTR) of the target mRNA. The seed sequence consists of 6–8 nucleotides at the 5′ end of the miRNA. In addition, it is thought that perfect complementary base pairing between the miRNA and mRNA results in rapid AGO-mediated endonucleotic cleavage degradation of the mRNA transcript [101]. On the other hand, partial complementarity between the miRNA:mRNA complex prevents the protein translation process and subsequently initiates cognate mRNA degradation. Messenger RNA degradation is initiated by de-adenylation facilitated by the interaction of a glycine-tryptophan protein (GW-182) and polyA-binding protein (PABP) [93,102]. GW-182 interacts with PABP to recruit two de-adenylases: CCR4 and CAF1 [93].

Although considered as highly stable cellular molecules, the degradation of miRNAs is regulated. Specific cell cycle stages [103,104], growth factor signaling [105], neuronal activity [106] and highly complementary mRNA targets [107] have all been shown to affect the stability and turnover of selected miRNAs. In contrast to mRNA, miRNAs have 5′ and 3′ unprotected ends rendering them accessible to exoribonuclease activity. Selective exonucleolytic decay of miRNAs is mediated by the 3′-5′ exoribonucleases XRN-1 along with ribosomal RNA-processing protein 41 (RRP41) [108]. Without affecting pri- or pre-miRs, the interferon-inducible 3′-to-5′ exoribonuclease human polynucleotide phosphorylase (hPNPase) degrades certain mature miRNAs [109]. Non-templated addition of adenosines to the 3′ end of mature mammalian miRNAs have both been observed to promote selective miRNA stability [110] and degradation [111], while modification of the 3′ end of either pre-miR or mature miRNA by the addition of uridines is implicated in miRNA degradation (Figure 2) [112].

### 4.2. Extracellular Roles of miRNA

Extracellular vesicle (EV)-mediated transfer of miRNAs has been shown to induce regulatory effects on recipient cells, thus mediating cell-to-cell communication [113]. From the International Society for Extracellular Vesicles position statement, the collective term EV covers not only membranous exosomes and microvesicles (MV), but also apoptotic bodies [114]. Of endocytic origin, exosomes (30–100 nm) are released into the extracellular environment, when multi-vesicular bodies (MVBs) fuse with the plasma membrane [115]. Inward invagination and detachment of MVBs containing multiple exosomes in the form of intraluminal vesicles (ILVs), gathering and sorting of EV cargo are mediated through the endosomal sorting complexes required for transport (ESCRTs) complexes 0, I, II and III [116]. Together with accessory proteins including tumor susceptibility gene 101 (TSG101) responsible for EGF-stimulated MVB formation and ALG2-interacting protein X (ALIX) which partakes in cargo packing and vesicle formation [117], ESCRTs incorporates selective miRNAs into exosomes. However, the cellular signals controlling this process is uncharacterized (Figure 3).

It is important that studies, describing beneficial effects on wound healing by exosomal miRNAs, appropriately report exosome classification. Often, studies fail to report enough specific molecular exosomal markers, including transmembrane (e.g., CD63, CD81) and cytosolic proteins (e.g., TSG101, ALIX) together with the absence of putative contaminants, including lipoproteins (e.g., APO, albumin); thus, the observed effects could merely be from biophysically similar EV and not exosomes.

## 5. microRNAs Altered in Diabetic Wound Healing

Among the many roles as regulators of gene expression, miRNAs have been demonstrated to mediate various skin pathologies such as persistent inflammation or excessive scar formation [118,119,120].

### 5.1. InflammiRs—microRNAs Having a Role in the Inflammatory Phase of Wound Healing

The inflammatory response in the wound is tightly regulated and any imbalance may cause chronic inflammation leading to poor healing. MiRNAs are very important molecules in the regulation of the inflammatory phase, and, here, we describe the role of several of them.

#### 5.1.1. MiR-132 Up-Regulation Decreases Pro-Inflammatory Responses

The level of miR-132 is highly up regulated in neutrophils following extravasation and infiltration into the skin [93]. Moreover, miR-132 was highly upregulated during the inflammatory phase of dermal wound repair, predominantly expressed in epidermal keratinocytes, and its expression continues to peak in the subsequent proliferative phase. MiR-132 has been shown to inhibit the expression of pro-inflammatory cytokines in several cells, such as keratinocytes, monocytes, and macrophages [121,122]. At the molecular level, miR-132 is induced by TGFβ1 and TGFβ2 in keratinocytes, and transcriptome analysis of keratinocytes revealed that miR-132 regulates a large number of immune response- and cell cycle-related genes [121]. Specifically, it down-regulates genes related to inflammatory pathways such as NFκB, NOD-like receptor, toll-like receptor, and TNF signaling pathways [123]. In keratinocytes, miR-132 decreased the production of chemokines and the capability to attract leukocytes by suppressing the NF-κB pathway. Conversely, miR-132 increased the activity of the that STAT3 and ERK pathways, thereby promoting keratinocyte growth [121]. Moreover, miR-132 can induce M2 polarization in macrophages, which is very important for successful wound repair [122].

Interestingly, it has been observed that miR-132 expression is significantly reduced in human diabetic ulcers compared with normal skin wounds, as well as in skin wounds of leptin receptor-deficient (db/db) diabetic mice compared with wild-type mice [124]. Restoring miR-132 in the wounds of db/db mice improved wound healing and topical application of liposome-formulated miR-132 mimic mixed with pluronic F-127 gel in a human ex vivo wound model also promoted wound re-epithelialization [124]. In addition to mir-132 anti-inflammatory potential by lowering the inflammatory cytokines a load, it has been also assigned the potential in mediating the transition from inflammation phase to proliferation (Table 2, Table 3 and Table 4) [77,122].

#### 5.1.2. MiR-146a Down-Regulation in Diabetic Wounds Results in Inflammation and Impaired Wound Healing

The levels of miR-146a are downregulated in diabetic mouse wounds. This miRNA is also associated with an increase in pro-inflammatory target genes, such as IL1 receptor-associated kinases 1 (IRAK1), tumor necrosis factor receptor-associated factor 6 (TRAF6), and NFκB, which results in increased levels of IL6 and MIP2 gene expression [136]. MiR-146a expression is upregulated by activation of the Toll-like receptors (TLR2-5), part of the innate immune response [138]. Moreover, miR-146a inhibits TLR2-induced inflammation, as a negative feedback loop, by targeting several key factors within the NFκB signaling pathway, such as IRAK1, IRAK2, and TRAF6 [138].

Furthermore, treatment with mesenchymal stem cells (MSCs) improved diabetic wound healing with a significant increase in the miR-146a expression level and a decreased gene expression of its pro-inflammatory target genes. While this evidence is indirect, these data suggest that the downregulated expression of miR-146a in response to injury in diabetic wounds could in part be responsible for the abnormal inflammatory response observed in diabetic wounds [136]. Moreover, the application of miR-146a conjugated with cerium oxide nanoparticles improved wound healing in diabetic mice without impairing the biomechanical properties of the healed skin (Table 2, Table 3 and Table 4) [139]. 

#### 5.1.3. MiR-21 Promotes M1 Macrophage Polarization

MiR-21 is upregulated in diabetic conditions [151] and mediates the hyperglycemic induction of NADPH oxidase 2 (Nox2) and thereby ROS production [152]. However, while skin miR-21 levels are highly up-regulated following wounding in normoglycemic mice, this increase is markedly blunted in diabetic mouse skin [151]. Thus, there is a clear dysregulation of miR-21 during diabetic wound healing. In line with these data, miR-21 was increased in fibroblasts obtained from DFUs compared with fibroblasts from normoglycemic subjects [153].

MiR-21 is highly expressed in macrophages and is involved in M1 macrophage polarization [154]. In the db/db mouse model and in wounds from diabetic patients, miR-21 expression was increased specifically in M1 macrophages [155]. In addition, an increase in miR-21 expression was shown in polarized M1 macrophages, and it was associated with an increased expression of the M1 markers IL1β, TNFα, iNOS, IL6 and IL8 [155]. Consequently, miR-21 was shown to inhibit the lipopolysaccharide (LPS)-induced inflammatory response and increase IL-10 production in macrophages [156]. Under diabetic conditions, miR-21 dysregulation may explain the abnormal inflammation and persistent M1 macrophage polarization observed in diabetic wounds, as miR-21 is over-expressed in the early phase of wound healing in diabetic skin but expressed at lower levels in the later phases [151]. Accordingly, overexpression of miR-21 in a wound healing rat model significantly improves the healing rate due to the activation of the AKT/PI3K signaling pathway by targeting the tumor suppressor phosphatase and tensin homologue (PTEN) (Table 2, Table 3 and Table 4) [157].

#### 5.1.4. MiR-155 Impairs Wound Re-Epithelization

Similar to miR-21, levels of miR-155 are not only increased in diabetic skin but fail to adequately decrease upon wounding under diabetic conditions [151]. Functionally, miR-155 is important for appropriate action of a multitude of skin cells involved in wound healing, including keratinocytes [169], dermal mesenchymal stem cells [172], mast cells [173], melanocytes [174], adipocytes [175] and fibroblasts [176]. Genetic absence of miR-155 improves wound healing [177], while acute miR-155 inhibition can restore wound healing in diabetic rodent models [178].

In immune cells, miR-155 has been shown to play a pro-inflammatory role via suppression of its targets cytotoxic T-lymphocyte-associated protein (CTLA)4 [179], suppressor of cytokine signaling (SOCS)1 [170], and SH2-Containing inositol-5′-phosphatase (SHIP) [170], which are all part of the Toll-Like Receptor (TLR)2 pathway [171]. Moreover, miR-155 also targets fibroblast growth factor (FGF7) (also known as Keratinocyte growth factor, KGF), and by decreasing FGF7, diabetes-induced miR-155 impairs re-epithelization, while miR-155 inhibition mediates de-repression of FGF7 and accelerates wound closure (Table 2, Table 3 and Table 4) [151].

#### 5.1.5. MiR-223 Is a Wound Inflammation-Related miRNA

Based on the available data, miR-223 is involved in the subtle control of different pathological conditions, ranging from inflammation [186,187,188] to infection [189] and cancer [190]. Neutrophils are an essential part of the innate immune response as they are critical for the first line of defense against bacteria and fungi. De Kerckhove and colleagues showed that miR-223 is expressed in neutrophils, controlled by the transcription factor c/EBPα and it represses IL6 production, therefore restraining neutrophil activation and allowing S. aureus growth in wounds [189]. Consequently, knock-down of miR-223 using anti-sense oligonucleotides improves wound-healing [189]. Furthermore, Zhou and colleagues demonstrated that miR-223 downregulates the canonical NFkB signaling by directly targeting its downstream targets Cul1a/b, Traf6, and Tab1 in basal epithelial cells [191]. This pathway is activated by cytokines such as IL1β, TNFα, or by LPS. In addition, it has been shown that the IKKα component is a target of miR-223 [192]. Similar findings were found by Matsui and Ogata working with periodontitis, a severe chronic inflammatory disease [193]. Human gingival fibroblasts (HGFs) stimulated with inflammatory cytokines (IL1β and IL6), after transfection with miR-223 mimic, have shown suppressed expression of the MKP-5 mRNA and IKKα protein [193]. Aside from being highly expressed in neutrophils present at the wound site and controlling neutrophil functions in acute inflammatory responses, miR-223 might also impact the recruitment of macrophages leading to a further chronic inflammatory response [58,187]. Macrophages infiltrate wound sites in the later inflammatory phase after neutrophil migration [62]. Of all inflammation-related miRNAs, miR-223 is predominantly expressed in neutrophils and macrophages at skin wound sites [189]. Macrophage numbers were significantly increased at the wound site in miR-223-deficient mice, at day 3 and 7 after injury. This indicates that miR-223 might regulate the acute inflammatory response at the wound site, subsequently affecting macrophage infiltration into wounds [189]. Furthermore, the over-expression of miR-223 facilitated M2 macrophage polarization in macrophages (RAW 246.5) through the inhibition of several genes of the glycolysis pathway and attenuated LPS-induced sepsis in a mouse model of acute liver failure (Table 2, Table 3 and Table 4) [194].

### 5.2. MiRNAs Affecting the Proliferative and Remodeling Phases of Diabetic Wound Healing

Restoring wound healing progression is dependent on a high manifestation of miRNAs at the later stages of wound healing. Some of the noteworthy and highly expressed miRNAs during these wound healing stages are mentioned here.

#### 5.2.1. Hypoxia-Regulated miR-210 Downregulates the Growth and Differentiation of Keratinocytes

‘HypoxymiRs’ denote miRNAs implicated in defining biological outcomes in response to changes in tissue oxygenation. Chronic non-healing wounds constitute a hypoxic environment, and miR- 210 as well as miR-21 are highly responsive to low oxygen levels in diabetic skin. Consequently, both miRNAs are increased in diabetic skin [151]. In hypoxia, the HIF1α transcription factor is stabilized to mediate the hypoxic response [125] encompassing the upregulation of miR-210. MiR-210 represses keratinocyte proliferation by targeting the cell-cycle regulatory protein E2F3 [126]. High levels of miR-210 at the wound-edges may therefore inhibit cell proliferation, a process necessary for healing. Accordingly, inhibition of miR-210 by intradermal injection of gramicidin liposome-based nanoparticles restores the epidermal cell proliferation capacity [127]. Furthermore, antagonizing the elevated miR-210, at the wound edges in a sulfur-mustard-induced wound healing model, promoted skin wound healing, partially through increasing keratinocyte migration (Table 2, Table 3 and Table 4) [128].

#### 5.2.2. The miR-200 Family Is Hypoxia-Sensitive and Triggers the Angiogenic Response

The miRNA-200 family, consisting of the miRNAs miR-200a, miR-200b, miR-200c, miR-429 and miR-141, plays an important role in mediating the vascularization of oxygen-insufficient tissues by repressing angiogenesis-related transcription factor Ets-1 and downstream genes, such as MMP2 and VEGF2 [140]. As opposed to miR-210, miR-200b is suppressed by extended hypoxia, allowing for the de-repression of Ets1 and the initiation of angiogenesis in the form of tube formation capacity [141]. Another important process necessary for successful wound healing is epithelial to mesenchymal transition (EMT), which is prerequisite for keratinocyte activation and their migration to the wound bed [142], and members of the miR-200 family have been already described as negative regulators of this transitory event [143]. MiR-200c has been found upregulated at the early phase of wound healing in aged skin. This implies that age-related increased levels of miR-200c in the wound epithelium could have a negative impact not only on keratinocyte migration but may also inhibit their differentiation [144]. Moreover, a similar re-epithelialization pathology has been described in chronic wounds (Table 2, Table 3 and Table 4) [195].

#### 5.2.3. Fibroblast Migration Is Dependent on miR-21 and miR-143/145

Even though miR-21 is up-regulated by inflammation, it is also important for the later stages of wound healing, when it is suppressed in diabetic skin [151,164]. This miRNA is enriched in the epidermis and hair follicles [165], which increases the migration of keratinocytes and improves re-epithelialization of the wound [166]. Thus, delivering miR-21 mimic via nanoparticles may stimulate re-epithelialization, an essential feature to ease healing and diminish scar formation [167]. Suppressing miR-21 at the later stages of wound healing leads to increased fibroblast migration [164]. Another study, conducted by the same group, has also shown that antagonizing miR-21 in the wound edge area significantly postponed the wound closure and disturbed the collagen bundle structure [168].

Similar to miR-21, both miR-143 and miR-145 are suppressed in the late stage of wound healing in diabetic skin [151]. This regulation allows for the activation of skin myofibroblasts and DFU-derived fibroblast migration and proliferation through stimulation of TGFβ1 and inhibition of α-SMA, necessary for facilitated wound contraction during wound healing [184]. MiR-145 has also been shown to target insulin receptor substrate-1 (IRS-1) and impairing insulin-mediated intracellular signaling in hepatic carcinoma [185]. It is also known to target PDGF D, which is the mitogen/chemoattractant for choroidal fibroblasts [153]. Therefore, suppression of miR-145 in the later stages of wound healing would therefore be expected to increase wound healing. However, these target genes were evaluated in other cells other than skin cells, and their relative importance as targets of miR-145 in skin may be different (Table 2, Table 3 and Table 4).

#### 5.2.4. MiR-29 Family and Collagen Deposition

The miR-29 family (miR-29a, miR-29b, and miR-29c) has an anti-fibrotic effect, mainly via its action on collagen, a highly conserved target of this family of miRNAs. These miRNAs are abundant in several tissues prone to fibrosis and they exert a range of cellular functions like apoptosis, proliferation, and differentiation [129]. MiR-29b is a potent post-transcriptional repressor of collagen 1 in skin fibroblasts and its deregulation might be implicated in scar formation [129]. The pharmacodynamic activity of an miR-29b mimic (remlarsen), which is in clinical trials, leads to the reversal of miR-29 expression and decreases scar formation via its effect on different collagens [130]. MiR-29b and miR-29c have been suggested as key mediators in the phenotypic switch from for scarless to scaring in the fetal wound gestational healing, during skin development [131], in part due to their roles in the extracellular matrix in fibrosis [132]. Both TGF-β, which acts anti-fibrotic in skin, and proteins such as SMADs, which are involved in the pathways for scar-free healing, are targets of miR-29b and miR-29c [133]. Moreover, biomechanical properties of diabetic skin tend to decline over time during the progression of the diabetic phenotype, and this decline results in decreased collagen content due to the dysregulation of miR-29a [134]. As suggested, the correction of these impairments is possible by both mesenchymal stem cell (MSC) treatment and the inhibition of miR-29a [134]. The topical application of miR-29b on the wound area in mice was found to increase the ratio of collagen types Ш to I and significantly increase the activity of metalloproteinase 8 and wound healing without a scar [135] whose transition is necessary for tissue remodeling (Table 2, Table 3 and Table 4) [35].

#### 5.2.5. MiR-126 Has a Role in Vascular Inflammation

Vascular tissues contain a high content of the miR-126 [158]. The peripheral blood of diabetic patients with ulcers carries lower levels of miR-126, than those without ulcers, and debridement therapy restores its levels [159]. Another study with both tissue and plasma obtained from diabetic rats undergoing accelerated wound closure by negative pressure wound therapy also supports the finding of upregulated miR-126 accelerated wound closure. A positive correlation between increased levels of circulating miR-126 and sprouting vascular network density was found. This was in part due to the inhibition of SPRED1 and PIK3R2, which are negative regulators of the VEGF receptor pathway [160]. Furthermore, endogenous miR-126 can inhibit the expression of vascular cell adhesion molecule-1 (VCAM1), which is a mediator in leukocyte trafficking in vascular inflammation [161]. Resting endothelial cells express miR-126, but do not transcribe mRNA of VCAM-1 to suppress inflammation in resting cells, while the VCAM-1 transcription is triggered by proinflammatory cytokines such as TNF-α and downstream transcription factors NFkB and IRF1 [162,163]. Targeted silencing of miR-126 in mice prompted a series of vascular defects in survivors, or caused lethality, through the increased expression of SPRED1, attenuating the activity of angiogenic signals via VEGF and FGF [158]. Exogenous delivery of hydrogel-encapsulated exosomes collected from synovial mesenchymal stem cells over-expressing miR-126-3p were used to treat wounds of diabetic animals, causing successful healing, and reinforcing re-epithelialization and promoting angiogenesis, via the MAPK/ERK and PI3K/AKT pathways (Table 2, Table 3 and Table 4) [163].

#### 5.2.6. MiR-198 Restrains Cell Proliferation

Follistatin-like 1 (FSTL1) is a secreted glycoprotein that has been involved in multiple signaling pathways, yet its role during diseases remains to be elucidated [145]. The FSTL1 primary transcript also encodes miRNA-198. During wound healing, the FSTL1/miR-198 transcript processing is altered by the KH-type splicing regulatory protein (KSRP) [146]. KSRP acts as a regulatory switch between miR-198 and FSTL1 to control keratinocyte migration and wound re-epithelialization. While FSTL-1 stimulated the migration of keratinocytes, miR-198 displayed an anti-migratory effect, and this reversible swap appears to be controlled by TGF-β signaling, by switching off miR-198 expression and promoting FSTL1 expression [146]. Thus, while FSTL1 promotes keratinocyte migration, miR-198 targets DIAPH1 (Diaphanous Related Formin 1), PLAU (Urokinase-Type Plasminogen Activator) and LAMC2 (Laminin Subunit Gamma 2) to impair migration. Moreover, this KSRP-dependent switch is impaired in DFUs; where miR-198 expression persists, keratinocyte migration and wound re-epithelialization are impaired [146]. Growing evidence supports the repressive action of miR-198 on the growth of several cancer cell types, including pancreatic [147] hepatocellular carcinoma [148] and osteosarcoma [149]. Moreover, miR-198 has been identified to target CCND2, which is a key regulator in cell cycle progression, and by binding to the 3′-UTR of CCND2 mRNA, miR-198 repressed CCND2 expression arresting the HaCaT cells in the G1 phase, thus inhibiting the proliferation (Table 2, Table 3 and Table 4) [150].

#### 5.2.7. The MiR-17∼92 Cluster Is Dysregulated in Ischemic Tissues

The miRNA17∼92 cluster, carrying the information about six individual miRNAs (miR-17, miR-18a, miR-19a, miR-19b, miR-20a, and miR-92a), has been validated as a vital player in biological functions of cell survival including differentiation, proliferation, angiogenesis and apoptosis [180]. The inhibition of miR-92a was previously shown as an effective strategy in recovering the vasculature network after tissue oxygen deprivation [181]. Another study shows that antagomiR-92, encapsulated with photosensitive protecting groups, efficiently brings down miR-92a expression levels, when excited with light at 385nm. This leads to de-repression of target genes Integrin 5α (Itga5) and Sirtuin 1 (Sirt1) in the wound, thereby raising proliferation and angiogenic response in diabetic skin [182]. Moreover, the miR-19 component of this cluster negatively controls WNT-signaling and angiogenesis to produce a denser vascular tree [180]. Furthermore, potentiating the miR-92a inhibition as a therapeutic strategy for chronic wounds is in the developmental stages, the locked nucleic acid (LNA)-modified miR-92a inhibitor, MRG-110, was confirmed to increase the expression of pro-angiogenic miR-92a target gene Itga5 in vitro and in vivo (Table 2, Table 3 and Table 4) [183].

## 6. Circulating and Tissue microRNAs as Biomarkers for the Early Detection of Diabetic Foot Ulceration

Prevention and early detection of wounds is crucial for achieving a reduction in amputations and complications associated with DFU. Differential expression of circulating and tissue miRNAs exhibit promising potential for diagnosis and treatment of DFU. However, it is important to evaluate whether and how novel miRNA findings are associated with relevant clinical outcomes. Moreover, intrinsic and extrinsic elements such as sex [196], age [197], type of biological sample [198], diabetes duration [199], method of profiling [200] as well as pre-analytic factor such as hemolysis of samples or covariates of DFU, such as decreased nutritional state of the patient, smoking, peripheral limb ischemia, etc., [201] may affect miRNA expression levels (Table 3).

**Table 3 cells-09-02228-t003:** Tissue and circulating microRNAs that occur in diabetic tissue, some of them are wound specific.

**Tissue specific microRNAs**				
**microRNA**	**Species**	**Expression pattern**	**Role in the tissue injury/repair**	**References**
miR-132	Humans Mouse	Reduced in human diabetic ulcers, upregulated in normal human wounds. Increased in diabetic mice wounds	Regulates the transition from inflammation to proliferation during wound healing	[121,124]
miR-146a	Human, Mouse	Increased in diabetic human skin, downregulated in mice wounds	Impairs inflammatory response observed in diabetic wounds	[136,138]
miR-21	Human, Mouse	Elevated in human DFUs and rodent skin	Inhibits the inflammatory response	[151,153]
miR-155	Human, Mouse	Increased in diabetic human and mice skin	Impairs wound re-epithelization	[151,178]
miR-223	Human, Mouse	Highly expressed in wound skin	Controls neutrophil functions in acute inflammatory responses	[189,194]
miR-210	Human, Mouse	High levels at the wound-edges	Impairs cell proliferative capacity	[127,128]
miR-200	Human	Suppressed in diabetic wounds	Mediates the vascularization and cell migration	[141,142]
miR-126	Human	Decreased in ulcers, increase after debridement therapy	Epithelialization and angiogenesis	[158,159]
miR-29	Human, Mouse	Increased in diabetic skin, decrease with wounding	Repressor of collagen 1 and scar formation	[129,130]
miR-203	Human	Increased in patients who underwent wound repairing surgery	Contributes to severity of the ulceration	[202]
**Circulating microRNAs**				
**microRNA**	**Species**	**Expression pattern**	**Role in the tissue injury/repair**	**References**
miR-217	Rat	Increased in serum	Regulation of the VEGF pathway through HIF-1α inhibition	[203]
miR-145	Human	Low in serum, higher in muscle from the amputated limbs	Fibroblast differentiation and suppression of cell growth	[184,204]
miR-16, -19b, -30e, -101, -144, -362, -451a, and -1260a	Human	Dysregulated in serum of patients with the Charcot foot (CF) and associated neuropathy	Monocytes differentiation	[205]
miR-191 and -200b	Human	Decreased in diabetes only, increased with chronic wounds and/or peripheral arterial disease (PAD)	Inflammation-mediated decrease of cellular migration and angiogenesis	[140]
miR-4739	Human	Increased in plasma and serum obtained from critical limb ischemia (CLI)	Critical limb ischemia including amputation	[206,207]
miR-129 and-335	Human	Skin and serum samples from patients with DFU	Impair wound healing due to elevated levels of MMP-9 together with transcription factor specificity protein 1 (Sp-1)	[208]

The known relative levels of specific miRNA in normal and diabetic wounds are also indicated, as well as a brief function of the listed miRNAs.

### 6.1. Circulating miRNAs Related to Diabetic Complications and Wound Healing

Local relative hypoxia plays a pivotal role in the normal wound healing process. HIF-1α, a known target of miR-217, regulates a series of genes including VEGF, as well as pathways involved in angiogenesis. Serum miR-217 is increased in diabetic patients, and even more so in diabetic patients suffering from DFUs [203]. When miR-127 inhibitors were applied to wounds, angiogenic activities were improved through the elevated HIF-1α expression and regulation of the VEGF pathway, in rodents [203].

MiRNA-145 is also known for its association with fibroblast differentiation [184] and suppression of cell growth [204]. Using serum from patients with DFU prior to amputation, an inverse correlation of low circulating serum miR-145 levels with high serum mRNA and protein levels of TGF-β1 was observed. Notably, the expression of miR-145 was higher in muscle from the amputated limb in these patients [209].

Charcot foot (CF) is a rare, although severe, complication associated with diabetes. Serum miRNA profiles were determined in T2D patients with acute CF, T2D with neuropathy but no CF, or T2D patients without CF and neuropathy. More than 51 differentially expressed circulating miRNAs were identified in T2D patients with CF. Using Ingenuity Pathway Analysis (IPA), eight interesting candidates (miR-16, -19b, -30e, -101, -144, -362, -451a, and miR-1260a) were found involved in monocytes differentiation; however, functional studies are needed to prove their role in the pathophysiology of the disease [205].

The involvement of plasma miRNAs in diabetes–associated impairment of tissue repair was investigated by Dangwal and colleagues [140]. They found that patients with T2D presented a decrease of up to 3-6-fold in circulating miR-191 and miR-200b while T2D patients with both peripheral arterial disease (PAD) and chronic wounds had elevated levels of miR-191 and miR-200b. Interestingly, the presence of PAD, but without chronic wounds, along with T2D had similar levels as T2D alone, thus indicating that PAD does not affect plasma miR-191 and miR-200b [140]. As described above, miR-200b is upregulated by hypoxia in skin, but additional factors may control its levels in circulation. Stimulating vascular endothelial cells with inflammatory cytokines, such as observed in T2D patients with chronic wounds, resulted in an inflammation-mediated increase in miR-191 and miR-200b [140]. Cellular migration and angiogenesis were inhibited upon transfection of dermal endothelial cells and fibroblast, grown under diabetic hyperglycemic conditions, with miR-191 or miR-200b [140].

The most severe manifestation of PAD is critical limb ischemia (CLI), which, in a cohort of Chinese T2D patients, was associated with elevated plasma miR-4739. Combining miR-4739 with known confounding risk factors such as T2D duration, hypertension, smoking, blood glucose and hyperlipidemia increased the diagnostic performance in receiver operator curves (ROC) analysis (AUC (area under the curve) = 0.94) compared to either miR-4739 alone (AUC = 0.69) or risk factors alone (AUC = 0.91) [206]. Another study also utilized serum samples obtained from CLI patients with and without T2D and found that not only were miR-15a and miR-16 increased in CLI subjects, but miR-15a levels specifically were also associated with adverse events, including amputation, at one year follow-up in T2DM patients who have undergone percutaneous angioplasty to treat CLI [207]. 

Non-healing, chronic DFUs present with elevated levels of MMP-9 together with transcription factor specificity protein 1 (Sp-1), which contribute to improper wound healing. Skin and serum samples isolated from patients with diabetic wounds, show [208] decreased miR-129 and miR-335 expression. Subsequently, Sp1-mediated regulation of the MMP-9 promoter activity through miR-129 and miR-335 has been demonstrated in HaCaT cells and primary keratinocytes. This suggests that local application of miR-129 and miR-335 may have therapeutic potential and promote diabetic wound healing (Table 3) [210].

### 6.2. MiRNAs in Diabetic Wounds as Possible Biomarkers for Wound Healing

Indexing and monitoring of DFUs using miRNAs could improve the sensitivity and specificity of the current DFU classification systems. Skin samples obtained from T2D individuals, who underwent wound repairing surgery, had increased expression levels of miR-203, which is a keratinocyte-specific miRNA highly abundant in epidermis [202]. Interestingly, the non-healing wounds had the highest expression level of miR-203 and the severity of the ulceration positively correlated with miR-203 expression [202].

Ramirez et al. (2015) investigated how diabetes, in general, affects intact human foot skin in order to identify predisposing features of DFU development [211]. They included laser-captured-microdissection (LCM) of skin samples from non-neuropathic T2D patients undergoing voluntary podiatric correction surgery. Surprisingly, only subtle changes to epidermal miRNAs were found, including the upregulation of miR-31 (both 3p and 5p). However, none of the miRNAs studied were statistically significant upon correction for multiple testing, not even in a set of prospectively collected skin samples (Table 3) [211].

### 6.3. Influence of Extracellular Vesicle miRNAs on Wound Healing

The miRNA profile of extracellular vesicles (EVs) from human clonal and primary keratinocytes is markedly different between different types of EVs—apoptotic bodies, exosomes and larger microvesicles. This is suggestive of active secretion of specific miRNAs into exosomes, possibly having roles in cell-to-cell communication [212]. In accordance, miRNAs in exosomes from amniotic epithelial cells or mesenchymal adipose-derived stem cells [213] or fibrocytes [214] can increase wound healing in vivo [215]. Moreover, exosomes from endothelial cells improve wound healing in diabetic animals by improving endothelial function [216]. EVs from diabetic patients are markedly different from non-diabetic controls with regard to their wound healing properties, cytokine, and RNA cargo, regardless of whether the EVs are isolated from plasma [217] or tissue keratocytes [218].

#### 6.3.1. miRNAs in Macrophage-Derived EVs

The molecular mechanisms by which EVs are loaded with RNA cargo are not well characterized, but specific miRNAs are differentially loaded into EVs from different cell types and have a functional impact on target cellular function. For example, miR-155 loaded macrophage-derived exosomes exacerbates cardiac dysfunction, suggesting that an miR-155 inhibitor can be used to protect cardiac function during infarction [219], supporting findings that a topical miR-155 inhibitor accelerates wound healing in a type 1 diabetes model [151]. Furthermore, the delivery of EVs, obtained from human umbilical cord blood mononuclear cells from different donors, was evaluated in three full-thickness excision wound models [220]. The pro-healing activity of these vesicles was noted by increased skin neovascularization and re-epithelization. These alterations were mainly induced by miRNA-150 that induced cell proliferation by repressing the c-MYB gene, thus identifying this miRNA as important in skin regeneration. 

#### 6.3.2. Endothelial Cell Derived EVs and Their miRNAs 

Endothelial microparticles (EMPs), which are EVs from endothelial cells, contain high levels of miR-126-3p. When endothelial cells were treated with EMPs containing miR-126, its target SPRED1, a known negative regulator of migration and proliferation, was inhibited. Interestingly, when EMPs were derived under diabetic conditions, levels of miR-126 were found 5.4 times lower, resulting in poor re-endothelialization, which in turn attenuated the regenerative capacity of the EMPs. Reduced miR-126 expression within circulating EMPs, but not from total plasma, of diabetic patients was also observed in the same study [221]. Diabetic mice treated with endothelial progenitor cell (EPC)-derived exosomal miR-221 also promoted angiogenesis and lead to accelerated wound closure [222].

However, whether EMPs also improves re-endothelialization in cutaneous wounds remains to be studied. Furthermore, whether the delivery by EVs is necessary for the effect of miR-126 or miR-221 remains to be studied.

Hyperbaric oxygen therapy (HBO) may, in some instances, improve neovascularization and thereby promote wound healing by increasing cell proliferation and viability as well as, endothelial tube formation. The proangiogenic endothelial-expressed lncRNA MALAT1 (metastasis-associated lung adenocarcinoma transcript 1) expression increased in human coronary artery endothelial cell (HCAEC)-derived exosomes following HBO treatment and suppressed levels of miR-92a leading to improved angiogenesis [223]. Adipose-derived stem cell (ADSCs) exosomes containing MALAT1 have also been shown to suppress miR-124 and thereby activate the WNT/β-catenin pathway leading to induced wound healing in a skin lesion model [224]. The protective role of MALAT1 delivered by exosomes is in contrast to the observed upregulation of cellular MALAT1 in various diabetes-related complications [225]. This indicates that something else other than MALAT1 is needed to enhance EV-mediated wound repair mechanisms. Yet, another exosomal-derived lncRNA H19 accelerated wound healing in mouse models of DFUs by suppressing miR-152-3p, thus resulting in an upregulation of PTEN [226].

#### 6.3.3. miRNAs Derived from MSC EVs Acting on Wound Healing

Transition from the inflammatory to the proliferative phase, aided by lipopolysaccharide (LPS)-primed MSCs show promising effects in orchestrating wound healing. Extracted exosomes from the supernatant of LPS-preconditioned MSCs (pre-MSCs) facilitated the production of anti-inflammatory cytokines. In addition, the M2 polarization in THP-1 cells, kept under a high-glucose environment, was mediated by the high expression of let-7b and the low expression of its target, TLR4. Alleviation of inflammation and enhanced cutaneous wound healing was also observed upon application of exosome-derived LPS pre-MSCs in streptozotocin-induced diabetic rats [227].

Furthermore, MSC derived EVs can enhance angiogenesis likely through their miRNA cargo. Controlled release of human exosomal synovial mesenchymal stem cell-derived miR-126 stimulated surface re-endothelialization and angiogenesis when applied as a wound dressing in diabetic rats [163]. Moreover, exosomal miR-125a, isolated from human adipose-derived MSCs, could be taken up by endothelial cells leading to enhanced angiogenesis both in vitro and in vivo by repressing angiogenic inhibitor delta-like 4 (DLL4) [228]. Conversely, over-expression of miR-125a-5p in human trophoblast cells resulted in decreased cell migration and inhibited angiogenesis through decreased VEGFA expression [229].

Circular RNAs can function as miRNA sponges. One example hereof is mmu_circ_0000250, which, when enclosed within adipose-derived MSCs exosomes and applied to EPC under high glucose conditions, could stimulate SIRT1 expression via the inhibition of miR-128. Moreover, exosomal-derived mmu_circ_0000250 restored angiogenetic activities and increased expression of autophagy-related proteins in EPCs under diabetic conditions [230].

#### 6.3.4. Circulating EV-Derived miRNAs and Their Actions in Wound Healing

Slow wound healing mediated by high levels of exosomal miR-20b, was observed in vivo and in vitro when exosomes obtained from T2D patients were injected into cutaneous mouse wounds or applied to HUVECs. Mechanistically, the miR-20b-mediated suppression of WNT9 led to impaired angiogenesis, while the silencing of miR-20b promoted wound healing in vivo [231]. Circulating exosomes, enriched with miR-15a and extracted from serum of diabetic patients, inhibited wound healing both in vitro an in vivo through the inhibition of the NOX5/ROS signaling pathway, while the inhibition of miR-15a improved wound closure, blood perfusion and angiogenesis [232].

Tissue miR-21, as described above, is important in several stages of cutaneous wound healing. Several studies have focused on circulating exosomal miR-21 [233], both the minor miR-21-3p [234] and the more prevalent miR-21-5p species [235,236,237]. In general, EVs with miR-21 improve wound healing through either a combination of enhanced fibroblast function, improved angiogenic activity and/or collagen remodeling by modulating important regulatory effector mRNAs such as MMPs, PTEN, SPRY1, WNT4, β-catenin, TIMP-1, TGF-β1 and THBS1. Although not all of the studies were performed in diabetic animals or under in vitro hyperglycemic conditions, the evidence is strong that miR-21 plays a pivotal role in wound healing and that this effect can be mediated by cell-to-cell communication via EVs (Table 3 and Table 4).

**Table 4 cells-09-02228-t004:** Actions of the various miRNA in wound healing.

Phases of Wound Healing	microRNA	Action in Wound Healing	Reference
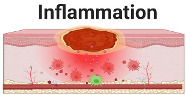	miR-132	Regulation of pro-inflammatory responses	[121,122,123,124]
	miR-146a	Inflammatory responses	[136,137,138,139]
	miR-21	Promotes M1 macrophage polarization	[151,152,153,154,155,156,157]
	miR-155	Wound re-epithelization	[151,169,170,171,172,173,174,175,176,177,178,179]
	miR-223	Inflammatory responses	[58,186,187,188,189,190,191,192,193,194]
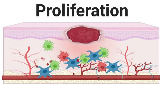	miR-210	Growth and differentiation of keratinocytes	[125,126,127,128]
	miR-200	Angiogenic response	[140,141,142,143,144]
	miR-126	Re-epithelialization and angiogenesis	[158,159,160,161,162,163]
	miR-17~92 cluster	Migration of keratinocytes and re-epithelialization of the wound	[180,181,182,183]
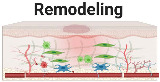	miR-29	Collagen restructuring and scar formation	[129,130,131,132,133,134,135]
	miR-198	Cell proliferation	[145,146,147,148,149,150]
	miR-21	Migration of keratinocytes and wound re-epithelialization	[164,165,166,167,168]
	miR-143/ miR-145	Cell differentiation, proliferation, angiogenesis and apoptosis	[184,185]

## 7. Conclusions and Perspectives

Some diabetic subjects appear to be protected from developing non-healing chronic foot wounds, while others are likely predisposed for the development of this devastating late-stage complication. As a result, there are two main points to consider of importance. One would be the discovery of specific molecular markers to be used for the early detection of this pathology, in subjects at a high risk of wound development. The other would be to discover novel targets for therapy to specifically address those individuals that will develop non-healing wounds. Molecules such as small noncoding RNAs, in particular miRNAs, important regulators of cellular functions, are good candidates to consider not only as early markers of disease, but they can also potentially be used as novel therapeutic agents to induce would healing and tissue repair in diabetic subjects.

So far, the field of early detection of high-risk individuals that are predisposed to developing foot wounds is not advanced enough to discriminate the individuals that will develop foot wounds from those that are protected, or to distinguish wounds that heal from non-healing wounds.

Much work still needs to be done in order to uncover early molecular alterations that may distinguish high-risk individuals for the development of chronic non-healing wounds. One possible and relatively novel mechanism of action, in this field, that could be used to evaluate and uncover non-healing wounds, before these are even formed, may be by epigenetic modifications of miRNA genes. Epigenetic changes, such as DNA methylation or histone modifications may be present long before non-healing wounds are detected and could be also be key as early targets of the disease if measured early on during disease development. Early alterations in these molecular pathways can contribute to physiological and pathological changes in persons with diabetes [238]. Epigenetic alterations are in part responsible for insulin resistance development and therefore they are associated with the development of related complications [14,239].

Moreover, miRNAs may also undergo adenosine-to-inosine editing by the ADAR1 or ADAR2 enzymes, and since inosine base pairs with cytidine rather than uridine RNA editing can change the target specificity of the miRNA. A large number of miRNAs have been shown to be edited in response to ischemia [240], but to what extent this occurs in diabetic wound healing has not been investigated.

Furthermore, the utilization of miRNAs as therapeutic targets is not only developing fast, but it is also gaining recognition. One important reason for this is that the expression of these molecules can be obtained not only in the blood of diabetic subjects, but their expression can also be related to that found in wound biofluids during the various stages of healing. The expression of these molecules may differ depending on the various stages of the pathology. The better we understand the molecular network of the pathways in each stage of wound healing, the better the therapeutic targets can be.

Both microRNA mimics, to enhance the expression of important microRNAs that are needed to induce healing, but are decreased under diabetic conditions, and antagomirs, which inhibit specific microRNAs that have been found highly expressed under diabetic conditions, can now be developed as therapeutic targets. These targets can be directed at very specific pathways that are known to be impaired under diabetes conditions and in particular during the various phases of wound healing. Therefore, a combination therapy, with several microRNAs, for topical applications may be more attractive since wound healing is complex and involves monitoring of both inflammation and infection.

## Figures and Tables

**Figure 1 cells-09-02228-f001:**
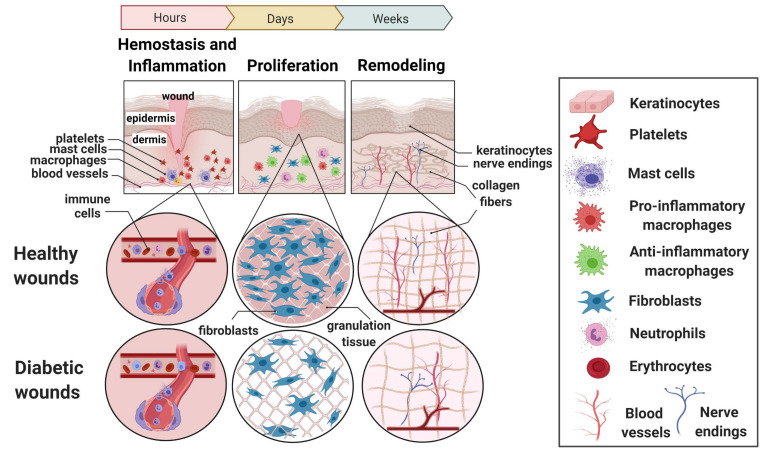
Three main stages of normal and diabetic wound healing. Also shown are the predominant skin cells involved at each stage of the process. Wound healing in healthy and diabetic skin is similar in the order of and key events occurring throughout the three overlapping phases. However, it differs significantly in the length and progression of the inflammatory phase under diabetic conditions, making the wounds fail through progression towards successful healing.

**Figure 2 cells-09-02228-f002:**
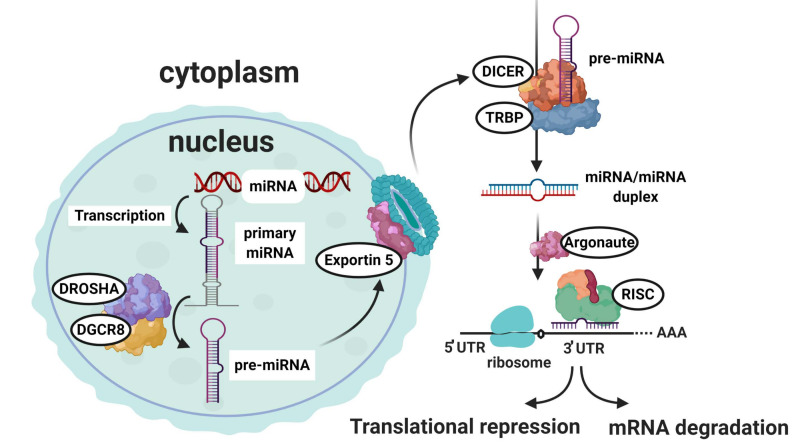
MicroRNA biogenesis and their predominant mode of action. The initial step in the microRNA (miRNA) biogenesis is generation of the primary (pri)-miR transcript. The DROSHA and DiGeorge Syndrome Critical Region 8 (DGCR8) microprocessor complex cleaves the pri-miRNA to form the precursor (pre)-miRNA. The pre-miRNA is then exported to the cytoplasm via Exportin 5. Exportin 5 delivers its cargo to be processed to the mature miRNA duplex by DICER. Finally, the mature miRNA duplex is loaded into the Argonaute (AGO) protein to form the miRNA-induced silencing complex (miRISC). In most cases, miRISC binds to target mRNAs to induce translational inhibition, but can also lead to mRNA degradation via recruitment of deadenylases.

**Figure 3 cells-09-02228-f003:**
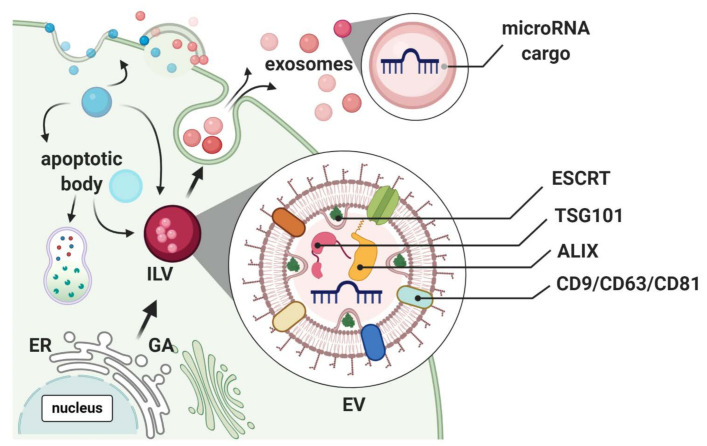
Extracellular vesicle (EV) generation and secretion. Molecules from extracellular space fuse with the plasma membrane, and reach the endosome, which buds inwardly to form intraluminal vesicles (ILV), which then transform into multi-vesicle bodies (MVBs). MVBs further fuse with the plasma membrane of *late endosomes* and release vesicles named exosomes into the outer space*.* Exosomes receive cargo through the endosomal sorting complex required for transport (ESCRT). Different types of exosomes can be discriminated not only by cargo type but also by different surface markers (CD63, CD9, CD81), or the tumor susceptibility gene 101 (TSG) 101 and the apoptosis-linked-gene-2 interacting protein X (ALIX).

**Table 1 cells-09-02228-t001:** Roles of the different cells and structures in the skin during wound healing.

Cell Type	Function	References
Keratinocytes 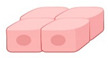	Under normal conditions, the main function of keratinocytes (KCs) is to form the protective barrier of the skin. As a response to the injury, keratinocytes secrete a vast yield of soluble fibrinogenic and angiogenic growth factors such as TGFα, TGFβ, VEGF, EGF, and KGF that prompt the regeneration in bordering tissue, during the wound healing process.	[38,39]
Platelets 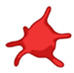	Platelets are one of the first cell types to respond to injury. They are a large source of growth factors like platelet-derived growth factor (PDGF) and transforming growth factor beta (TGF-β) in the early wound.	[40,41]
Dendritic cells 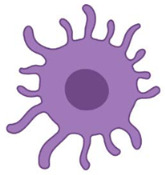	Dendritic cells (DCs) are antigen-presenting cells also known as messengers between the innate and the adaptive mammalian immune response during each phase of wound healing.	[42,43]
Neutrophils 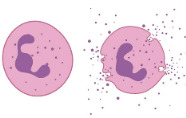	Neutrophils are in the first line of response of the innate immune response, producing an immediate and potent response against invading and harmful agents. Neutrophils are highly motile and abundantly recruited and amplified in response to cytokines released from damaged and necrotic cells after tissue injury.	[44,45]
Endothelial cells 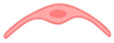	Endothelial cells (ECs) respond to the wound healing signals released by keratinocytes and fibroblasts, initiating angiogenesis, where they have a special role in augmenting the growth and survival of newly formed tissue.	[46]
Myofibroblasts and fibroblasts 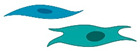	Myofibroblasts are typically activated fibroblasts that contribute to wound healing by generating extracellular matrix. They also propagate a contractile strength to the wound edges during wound contraction. Myofibroblasts differentiation is frequently induced by endothelin-1, TGFβ, and cellular fibronectin. Fibroblasts are one of the key players in the wound contraction related events like resolving the fibrin clot, and establishing the extra cellular matrix (ECM) and collagen deposition.	[47]
Lymphocytes 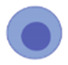	Lymphocytes play an important regulatory role in wound healing and scar formation.	[48]
Macrophages 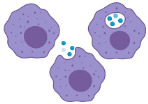	Two fractions present in the tissues. Tissue resident macrophages are constitutively present, while the recruited macrophages appear at the wound site and play an important role in clearing the matrix, cell debris and microorganisms. They coordinate tissue repair.	[49,50]
NK-cells 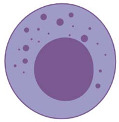	Natural killer (NK) cells regulate the inflammatory phase in wound repair, as well as the later stages of wound healing (re-epithelialization, angiogenesis, granulation tissue formation, and the remodeling phase).	[51]
B-cells 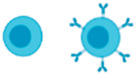	B cells are part of the humoral branch of the immune system. One of the studies shows that the infiltration of B-cells into the wound is mediated through interleukin 10 (IL-10).	[52]
T-cells 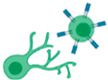	T-cells localized at the epidermis and may play regulatory roles in skin tissue homeostasis and repair.	[53,54]
Cutaneous innervation 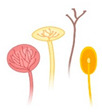	Skin physiological and pathophysiological properties rely on the sensory neuronal network (consisting of various epidermal and dermal receptors; and sensory autonomic nerve fibers). Sensory innervation coordinates skin responses by releasing different kinds of neuropeptides, which in turn activate skin cells during the immune response.	[55,56,57,58]

**Table 2 cells-09-02228-t002:** microRNAs involved in different stages of wound healing.

Wound Healing								
Inflammation			Proliferation			Remodeling		
microRNA	Target	Reference	microRNA	Target	Reference	microRNA	Target	Reference
miR-132	NFκB, TNFα, STAT3, IRAK4	[121,122,123,124]	miR-210	E2F3	[125,126,127,128]	miR-29	COL 1-3, TGF1β, SMAD	[129,130,131,132,133,134,135]
miR-146a	TRAF6, IRAK1, IRAK2	[136,137,138,139]	miR-200	MMP2, VEGF2 ZEB1 SIP1	[140,141,142,143,144]	miR-198	FSTL1, CCND2	[145,146,147,148,149,150]
miR-21	NOX2, IL1β, iNOS, IL6, TNFα, PTEN	[151,152,153,154,155,156,157]	miR-126	SPRED1, PIK3R2, VCAM1	[158,159,160,161,162,163]	miR-21	TGF1β	[164,165,166,167,168]
miR-155	TLA4, SOCS1, SHIP, FGF7	[151,169,170,171,172,173,174,175,176,177,178,179]	miR-17~92	TGF1β, SMAD1, ITGA5	[180,181,182,183]	miR-143/miR-145	IRS1, PDGFD, αSMA	[[184],[185]
miR-223	MKP5, IKKα	[58,186,187,188,189,190,191,192,193,194]						

The figure shows highly expressed microRNAs in the skin during the inflammatory, proliferative and remodeling phases of wound healing and their known targets in coordinating different process in the healing skin.

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
