# Peer review of "Mechanistic Actions of microRNAs in Diabetic Wound Healing"

_cells, 2020, doi:10.3390/cells9102228_

Round 1
Reviewer 1 Report
The present manuscript is a well written and interesting review on wound healing in diabetes. The text begins with an accurate description of the normal wound healing phases. It then proceeds to a description of the events in diabetic foot ulceration (DFU) and, finally, the role of miRNAs.
Maybe too much emphasis was given to DFU instead of non-healing wounds in general.
Furthermore, section 4, which explains the biogenesis and transport of miRNAs, as well as Figure 2, are not necessary. These pieces of information are all easily found in textbooks of Biochemistry and Molecular Biology.
Conversely, the actions of the various miRNA cited should be summarized in a figure, in “Conclusion”.
Author Response
The present manuscript is a well written and interesting review on wound healing in diabetes. The text begins with an accurate description of the normal wound healing phases. It then proceeds to a description of the events in diabetic foot ulceration (DFU) and, finally, the role of miRNAs.
Point 1: Maybe too much emphasis was given to DFU instead of non-healing wounds in general.
Response 1: We would like to thank you for taking the time to assess our manuscript. With no intention to underestimate the importance of the other wound etiologies like venous and pressure ulcer, our intention with this review was to give an insight on the chronicity of the wounds using the diabetic foot ulcers as the most represented chronic wounds. However, we found your comment extremely useful and modified the introductory part so that we address this accordingly (line 29-37).
Point 2: Furthermore, section 4, which explains the biogenesis and transport of miRNAs, as well as Figure 2, are not necessary. These pieces of information are all easily found in textbooks of Biochemistry and Molecular Biology.
Response 2: Thank you! We have shortened significantly the description of the microRNA biogenesis rather than completely removing it. We believe this set the information out very briefly clearly and is a format that readers, especially audience like students will readily return to when seeking on microRNA biogenesis information.
Point 3: Conversely, the actions of the various miRNA cited should be summarized in a figure, in “Conclusion”.
Response 3: Thank you for your suggestion, which we appreciate. The various miRNA described in this review are now summarized in the table format in the conclusion part (table 4).
Reviewer 2 Report
This review focuses on the role of microRNA in the context of impaired healing of diabetes, which is a topic of emerging interest. The publication of a review on this topic would be a welcome addition to the literature. In general, the review is clearly written, but there are a few gaps that need to be filled. Suggestions are also made to improve clarity and organization.
- The final statement of the abstract is not really in keeping with the review, as the review considers biomarkers, but primarily focuses on the function of miRNAs in diabetic wounds.
- Table 1 should be revised/reconsidered. Many of the cell types that are listed are not mentioned at all in the portion of the review that discusses wound healing (although they probably should be, see comment #3). Moreover, the descriptions that are provided in the Table are uneven and therefore are not uniformly useful. For example, no information is provided in the Table about the actual function of neutrophils or B cells in wounds, nor is there any direct information about the true role of keratinocytes other than that they “respond to injury”. The role of innervation should probably be added to the table as well, given the issues that occur in diabetes. Fibroblasts are another prominent omission. Finally, since Table 1 seems to relate to skin, the table title should include “in skin” or “in epithelium”.
- The beginning part of the review is very macrophage focused. Although macrophage dysfunction is one of the best-studied deficiencies in DFUs, it is far from the only one. For example, neutrophil dysfunction has been demonstrated to be important in impaired diabetic healing, along with neuropathy, endothelial, and epithelial dysfunction, to name a few. Many of these become important in the later section on miRNA dysfunction in diabetes.
- The sentence that begins on line 73 is a bit of an overstatement. While there is a great deal of evidence for a role for macrophages in wound healing, evidence for lymphocytes is more scarce. In addition, lymphocytes are not further considered in this review, leaving a question as to the intent of mentioning them in the opening sentence of this paragraph.
- Section 3.2 might be better organized if it begins with the normal polarization pattern, and then indicates how the phenotypic shift is aberrant in diabetic wounds, rather than the other way around.
- Table 2 seems quite valuable to the reader, but the text does not refer to it. What seems to be missing, though, is an additional Table that catalogues the differences in miRs that occur in diabetic wounds. This type of summary of the text would be very useful, as its quite hard to synthesize the lengthy text that describes the large number of miRs changes in diabetic wounds. The text that marches through the functional aspects of the various miRs that play a role in diabetic dysfunction is okay, but it’s hard for the average reader to get a clear picture after reading it, and the specialized information may not be commonly useful. The addition of a table would help the general reader, as it could summarize what is known about relative levels of specific miRNA in normal and diabetic wounds, indicate whether data is human, animal, or both, and provide a brief function of the miRNA. A table of possible biomarker miRNA for non-healing DFU would also be useful.
- The authors should consider adding a figure describing the type and effects of EV association miRNAs.
Author Response
This review focuses on the role of microRNA in the context of impaired healing of diabetes, which is a topic of emerging interest. The publication of a review on this topic would be a welcome addition to the literature. In general, the review is clearly written, but there are a few gaps that need to be filled. Suggestions are also made to improve clarity and organization.
Point 1: The final statement of the abstract is not really in keeping with the review, as the review considers biomarkers, but primarily focuses on the function of miRNAs in diabetic wounds.
Response 1: Thank you for your assessment of our manuscript. We have modified the final statement in the abstract so that our message is more accurate (lines 17-20).
Point 2: Table 1 should be revised/reconsidered. Many of the cell types that are listed are not mentioned at all in the portion of the review that discusses wound healing (although they probably should be, see comment #3). Moreover, the descriptions that are provided in the Table are uneven and therefore are not uniformly useful. For example, no information is provided in the Table about the actual function of neutrophils or B cells in wounds, nor is there any direct information about the true role of keratinocytes other than that they “respond to injury”. The role of innervation should probably be added to the table as well, given the issues that occur in diabetes. Fibroblasts are another prominent omission. Finally, since Table 1 seems to relate to skin, the table title should include “in skin” or “in epithelium”.
Response 2: Thank you so much for catching these confusing omissions, which we have now corrected. Each changed word is marked in yellow in the revised paper, and we hope these alterations give the better overview of the cellular players in wound healing (Table 1).
Point 3: The beginning part of the review is very macrophage focused. Although macrophage dysfunction is one of the best-studied deficiencies in DFUs, it is far from the only one. For example, neutrophil dysfunction has been demonstrated to be important in impaired diabetic healing, along with neuropathy, endothelial, and epithelial dysfunction, to name a few. Many of these become important in the later section on miRNA dysfunction in diabetes.
Response 3: We agree with your observation. We have gone through the entire manuscript carefully and adjusted relevant parts so that we display a more balanced view of the relations between the cell dysfunctions-related impairments in wound healing (lines 97-105).
Point 4: The sentence that begins on line 73 is a bit of an overstatement. While there is a great deal of evidence for a role for macrophages in wound healing, evidence for lymphocytes is more scarce. In addition, lymphocytes are not further considered in this review, leaving a question as to the intent of mentioning them in the opening sentence of this paragraph.
Response 4: Thank you for this excellent observation. We have replaced this statement of the importance of the macrophages with a moderated sentence also emphasizing the role of lymphocytes (line 80).
Point 5: Section 3.2 might be better organized if it begins with the normal polarization pattern, and then indicates how the phenotypic shift is aberrant in diabetic wounds, rather than the other way around.
Response 5: We agree and we have reorganized the text accordingly (lines 163-189).
Point 6: Table 2 seems quite valuable to the reader, but the text does not refer to it. What seems to be missing, though, is an additional Table that catalogues the differences in miRs that occur in diabetic wounds. This type of summary of the text would be very useful, as it is quite hard to synthesize the lengthy text that describes the large number of miRs changes in diabetic wounds. The text that marches through the functional aspects of the various miRs that play a role in diabetic dysfunction is okay, but it’s hard for the average reader to get a clear picture after reading it, and the specialized information may not be commonly useful. The addition of a table would help the general reader, as it could summarize what is known about relative levels of specific miRNA in normal and diabetic wounds, indicate whether data is human, animal, or both, and provide a brief function of the miRNA. A table of possible biomarker miRNA for non-healing DFU would also be useful.
Response 6: Thank you for reminding us how important it is to present complex material like this in the visual form. We have prepared a table to provide a synthesis of this information (Table 3).
Point 7: The authors should consider adding a figure describing the type and effects of EV association miRNAs.
Response 7: We completely agree and we have done so (Figure 3).
Reviewer 3 Report
With their manuscript ‘Mechanistic actions of microRNAs in diabetic wound healing’, the authors presented a summary of various articles (total: 225) about the emerging role of miRNAs in wound healing. Chronically healing wounds are serious medical problems due to their common and frequent complications. Thus, it is always very welcome to review factors contributing to understanding mechanisms responsible for delayed diabetic wound healing.
The article is well written and text is well organized. However, it should be noted that in addition to an improved understanding of miRNA functions, epigenetic control mechanisms are becoming more apparent in the discussed field. Metformin, for example, is capable of regulating miR-221, miR-222, and miR-34a. Moreover, the metformin-induced activation of DICER was also described. Therefofe, a detailed explanation of epigenetic regulation should be added
Author Response
With their manuscript ‘Mechanistic actions of microRNAs in diabetic wound healing’, the authors presented a summary of various articles (total: 225) about the emerging role of miRNAs in wound healing. Chronically healing wounds are serious medical problems due to their common and frequent complications. Thus, it is always very welcome to review factors contributing to understanding mechanisms responsible for delayed diabetic wound healing.
Point 1: The article is well written and text is well organized. However, it should be noted that in addition to an improved understanding of miRNA functions, epigenetic control mechanisms are becoming more apparent in the discussed field. Metformin, for example, is capable of regulating miR-221, miR-222, and miR-34a. Moreover, the metformin-induced activation of DICER was also described. Therefore, a detailed explanation of epigenetic regulation should be added.
Response 1: Thank you for the comment! While we agree that epigenetic modifications are likely to play a major role in the regulatory landscape of diabetic wound healing, we also feel that a thorough coverage of these mechanisms would make this review too extensive. Therefore, we have restricted ourselves to only include epigenetics of microRNA regulations in the discussion section of the revised manuscript (lines 63-65 in the introduction and lines 702-720 in the conclusion).
Round 2
Reviewer 3 Report
I am generally satisfied with the revisions, and I recommend the acceptance of this work.